# The Immunomodulator Dimethyl Itaconate Inhibits Several Key Steps of Angiogenesis in Cultured Endothelial Cells

**DOI:** 10.3390/ijms232415972

**Published:** 2022-12-15

**Authors:** Isabel Vidal, Elena Fernández-Florido, Ana Dácil Marrero, Laura Castilla, Ana R. Quesada, Beatriz Martínez-Poveda, Miguel Ángel Medina

**Affiliations:** 1Departamento de Biología Molecular y Bioquímica, Facultad de Ciencias, Universidad de Málaga, Andalucía Tech, E-29071 Málaga, Spain; 2Instituto de Investigación Biomédica de Málaga y Plataforma en Nanomedicina—IBIMA-Plataforma BIONAND, E-29071 Málaga, Spain; 3CIBER de Enfermedades Raras (CIBERER, Instituto de Salud Carlos III), E-29071 Madrid, Spain; 4CIBER de Enfermedades Cardiovasculares (CIBERCV, Instituto de Salud Carlos III), E-29071 Madrid, Spain

**Keywords:** dimethyl itaconate, angiogenesis, endothelial cells

## Abstract

The dimethyl derivative of the immunomodulator itaconate has been previously shown to have anti-inflammatory, anti-oxidative, and immunomodulatory effects. In the present work, we evaluate the potential of dimethyl itaconate as an anti-angiogenic compound by using cultured endothelial cells and several in vitro assays that simulate key steps of the angiogenic process, including endothelial cell proliferation, migration, invasion, and tube formation. Our results show that dimethyl itaconate interferes with all the previously mentioned steps of the angiogenic process, suggesting that dimethyl itaconate behaves as an anti-angiogenic compound.

## 1. Introduction

Itaconate is a known immunomodulator metabolite synthesized from the Krebs cycle intermediate cis-aconitate by the enzyme aconitate dehydrogenase 1 [1,2,3,4]. This is the initial member of a growing family of immunomodulators [5].

Cell-permeable derivatives, such as 4-octyl itaconate and dimethyl itaconate (DMI, see Figure 1), are commercially available and are useful for studying the biological effects of itaconate. However, it has been reported that DMI is not metabolized into itaconate intracellularly [6]; hence, the biological effects of DMI should be considered as caused by this compound, despite its tight connections with itaconate.

The main immunomodulatory, anti-inflammatory and anti-oxidative effects described so far for DMI are summarized in Figure 1. These documented biological effects of DMI will be further commented on in the discussion section. Some of these effects indirectly point to the potential effects of DMI on (lymph)angiogenesis. However, so far, nothing has been published regarding the connections of DMI with angiogenesis. The aim of the present work is to evaluate the potential interfering effects of DMI on the key steps of angiogenesis in cultured endothelial cells.

## 2. Results

### 2.1. DMI Inhibtis Endothelial Cell Growth

In adults, endothelial cells remain essentially quiescent. However, when the angiogenic phenotype is switched on, endothelial cells are proliferative. Endothelial cell proliferation can be easily mimicked in culture. Under proliferative conditions, cultured endothelial cell growth is expected to be affected by treatments with cytotoxic or cytostatic effects. The inhibition of endothelial cell proliferation can be considered a primary effect of anti-angiogenic compounds [7,8].

Figure 2A shows that DMI treatment drastically affects bovine aortic endothelial cells BAEC) growth in the submillimolar range of concentrations, with an IC_50_ estimated value of 276 ± 53 µM. Figure 2B shows that a similar trend can be observed for the DMI treatment of human umbilical endothelial cells (HUVEC). In this case, the estimated IC_50_ value is 491 ± 76 µM. From here on, the rest of the experiments were carried out using BAEC and DMI concentrations around the IC_50_ value for BAEC. Figure 2C shows that, indeed, DMI not only affects the overall growth of endothelial cells but has inhibitory effects on BAEC proliferation at 250 and 500 µM, as detected with the EdU flow cytometry assay.

### 2.2. DMI Inhibits Endothelial Cell Tubular Structure Formation on Matrigel

It is possible to evaluate whether DMI can inhibit the morphogenesis of tubular structures, a key final step of the angiogenic process, by using the tubule formation assay on Matrigel.

The DMI concentrations selected for the tubulogenesis assay were 500, 250, and 125 µM. As shown in Figure 3A, which includes representative images of each of the treatments that were carried out, a marked reduction in the formation of tubular structures at 500 µM DMI could be observed with the naked eye; however, the reduction was less noticeable for in case of 250 µM DMI. For the lowest concentration tested, 125 µM DMI, there was no apparent change in the tube formation. The quantification of the number of tubes formed in each of the treatments and subsequent statistical analysis (Figure 3B) support the concept that DMI significantly inhibits the formation of tubular structures at 500 and 250 µM, as the percentage of tubes formed was relative to the negative control being 33% and 63%, respectively. On the other hand, 83% of the tubes were formed in the presence of 125 µM DMI, with no significant differences with respect to the negative control of the experiment.

### 2.3. DMI Does Not Affect the Endothelial Cell Population Distribution along the Different Phases of Cell Cycle

Simple propidium iodide staining allows for the separation of the cell population in the different phases of the cell cycle using the flow cytometry assay as described in the material and methods section. Figure 4 shows that DMI at the tested concentrations had no relevant effect on the BAEC cell cycle. The obtained result suggests that any pro-apoptotic effect of DMI on BAEC could be discarded.

### 2.4. DMI Inhibits Endothelial Cell Migration

To evaluate the effect of DMI on cell migration, a wound healing assay was performed to quantify the reoccupied area of the initial wound after 4 and 8 h of incubation (Figure 5). Four hours after the wound was made, the percentage of the area recovered in the control was approximately 40%, while in the presence of 500 µM DMI, only 22% of the initial total area was reoccupied. A higher value of reoccupation was obtained with 250 and 125 µM DMI. After eight hours of incubation from wounding, 67% of the area was reoccupied in the control wells, while in the wells treated with 500, 250, and 125 µM DMI, 25%, 44%, and 55% reoccupation with respect to the initial wound area was obtained. The statistical analysis of these results supports the concept that 500 µM DMI significantly reduces the migration of the BAEC cell culture after both 4 and 8 h of wounding.

### 2.5. DMI Inhibits Endothelial Cell Invasion but Does Not Affect Matrix Metalloproteinase-2 Secretion

To progress in angiogenesis, endothelial cells should not only migrate but also invade the tissues to be vascularized. By using the transwell invasion assay described in the material and methods section, we were able to show that DMI strongly inhibits the invasive potential of BAEC at all the tested concentrations (Figure 6A,B). The statistical analysis of the obtained results revealed that this inhibitory effect was highly significant even at the lowest DMI concentration tested.

This invasive potential of endothelial cells during angiogenesis was achieved through the activation of the production and secretion of the extracellular matrix remodeling enzymes and/or by inducing a change in the balance of these enzymes’ activities and those of their specific inhibitors. For pathological angiogenesis, the role of matrix metalloproteinase 2 (MMP-2) secreted by endothelial cells was especially remarkable. In our hands, the gelatin zymography of BAEC-conditioned media revealed no relevant change in the secreted MMP-2 (Figure 6C).

## 3. Discussion

DMI is a membrane-permeable derivative of the immunomodulatory metabolite itaconate [1,2,3,4]. Figure 1 summarizes the main biological effects described for DMI so far. DMI, due to its structural properties, acts as an electrophile and is, therefore, capable of inducing electrophilic stress. At the cellular level, this stress affects GSH reserves, inducing an increase in reactive oxygen species (ROS) inside the cell. This makes it a very potent Nrf2 activator, superior to itaconate, and similar to dimethyl fumarate (DMF), thus producing the activation of both Nrf2-dependent and Nrf2-independent oxidative stress responses [11,12]. The activation of Nrf2 by DMI leads to the increased expression of Nrf2 transcription factor target genes, including heme oxygenase 1 (Hmox1).

Another target of DMI is the NF-kB signaling pathway. In concrete, it has been shown that the production of TNF-α and IL-1β and the phosphorylation of p65 NF-kB are reduced by DMI treatment in a mouse model of mastitis [13].

DMI has also been shown to have good results for the treatment of psoriasis in animal models of the disease due to the selective inhibition of the IL-17-IκBζ pathway involved in skin pathology [12]. Moderate to severe psoriasis has been treated in Germany for more than 50 years with DMF [14], the Nrf2 activator that has become the election treatment for multiple sclerosis [15,16,17]. Moreover, DMI not only inhibits the induction of IκBζ but also inhibits the activation of pro-IL1β as well as the secretion of IL-6, IL-12, IL-10, and interferon-β in an Nrf2-independent manner [18].

More recently, it has been shown that DMI inhibits pyroptosis through the inhibition and activation of canonical NLRP3 inflammasome [19,20].

The relevant anti-inflammatory role of DMI has raised interest in its potential application in the treatment of other diseases, such as autoimmune encephalomyelitis. A recent study has identified that DMI reduces blood–brain barrier damage in experimental mice with autoimmune encephalomyelitis, possibly due to the inhibition of the activity of metalloproteinases 3 and 9 [21]. The protective effects of DMI have also been described for endometritis, colitis-associated colorectal cancer, sepsis, and hepatocellular carcinoma, among others [22,23,24,25]. These protective effects were determined in vivo, showing that the daily administration of DMI had no significant effect on the succinate dehydrogenase activity of the heart and liver, thus suggesting its lack of toxicity [12]. DMI even increased the survival rate in mice treated with LPS to induce a sepsis shock in contrast with the PBS-treated population; however, this effect was not evident in Nrf2 ^−/−^ mice [24]. Furthermore, mice with ulcerative colitis induced by dextran sodium sulfate were shown to maintain their weight after 8–9 days of treatment with DMI in comparison with the untreated population [23].

Several of the described biological effects of DMI can be connected to angiogenesis: (1) as mentioned above, DMI could be useful for psoriasis treatment, as is the case for DMF [14]. Our group and another group independently were the first to demonstrate that this anti-psoriatic effect of DMF could be due to its anti-angiogenic activity [26,27]. (2) The NF-kB signaling pathway is connected with angiogenesis [28,29]. As mentioned above, it has been shown that the NF-kB signaling pathway is targeted by DMI [13]. (3) DMI inhibits the release of IL-6, a pleiotropic cytokine that regulates the immune response and hematopoiesis. This interleukin is an inducer of lymphangiogenesis and the formation of endothelial tubular structures within the tumor [30]. Specifically, IL-6 is released by cells in the tumor microenvironment and increases the production and subsequent release of VEGF into the environment via the IL-6/STAT3/VEGFA pathway, promoting endothelial cell activation and the subsequent formation of tubular structures [31]. This effect of IL-6 on angiogenesis has also been demonstrated in vitro in human endothelial cells, where the binding of IL-6, secreted in an autocrine manner, to the IL-6/gp130 receptor present on endothelial cells is essential to maintaining the angiogenic function. Indeed, a reduction in IL-6 expression in endothelial cells resulted in impaired tube formation by endothelial cells [32].

Based on the aforementioned scientific information, it seems reasonable to enunciate the following hypothesis: *DMI could interfere with the key steps of angiogenesis.* The results obtained in the present work validate this working hypothesis since we have proven that DMI inhibits endothelial cell growth, as well as migration, invasion, and tube formation: all of them critical steps of the angiogenic process. In fact, we have shown that DMI inhibits both BAEC and HUVEC growth in a dose-dependent manner with IC_50_ values in the submillimolar range of concentrations. Although the IC_50_ value in BAEC treated with DMI is higher than that previously reported for DMF [26], it is similar to those values reported for other compounds previously described as anti-angiogenic [33,34]. Both the known biological properties of DMI and the results obtained with DMI in the tubule formation assay on Matrigel, the wound healing assay, and the invasion assay, suggest that DMI could be considered a new anti-angiogenic compound with similar features to those previously shown for DMF [26]. However, DMI as an anti-angiogenic compound exhibits some differential features; for instance, in high contrast to DMF, DMI does not alter the BAEC cell cycle and does not seem to induce BAEC apoptosis. The potential future application of DMI for the treatment of angiogenesis-dependent diseases deserves to be further studied and warrants future experimental efforts, including in vivo studies.

## 4. Materials and Methods

### 4.1. Materials

Dulbecco’s modified Eagle’s medium (DMEM) containing glucose (1 g/L and 4.5 g/L) was purchased from Corning (Corning, NY, USA), while an endothelial cell growth medium-2 (EGM-2) was obtained from LONZA (Muenchensteinerstrasse, Switzerland). Penicillin, streptomycin, and trypsin were purchased from BioWhittaker (Verviers, Belgium). Fetal bovine serum (FBS) was obtained from BioWest (Nuaillé, France). Acrylamide was purchased from BioRad (Hercules, CA, USA). Matrigel was purchased from Becton-Dickinson (Bedford, MA, USA). Plastic material for cell culture was obtained from Nunc (Roskilde, Denmark). All other reagents not listed in this section were purchased from Sigma-Aldrich (St. Louis, MO, USA).

### 4.2. Cell Culture

Cell culture media were supplemented with penicillin (50 U/mL) and streptomycin (50 U/mL). Bovine aortic endothelial cells (BAEC) were isolated from bovine aortic arches as previously described [35] and maintained in DMEM containing glucose (1 g/L) supplemented with 10% FBS and glutamine (2 mM). Human umbilical endothelial cells (HUVEC) were obtained from Lonza and maintained in an EGM-2 medium (Lonza); HUVECs were used until pass 9 following the manufacturers’ instructions. The human osteosarcoma U2OS cell line was maintained in DMEM containing glucose (4.5 g/L) supplemented with 10% FBS and glutamine (2 mM). All cell lines were maintained at 37 °C under a 5% ${\mathrm{CO}}_{2}$ humidified atmosphere.

### 4.3. Cell Survival Assay under Proliferative Conditions (MTT Assay)

The 3-(4,5-dimethylthiazol-2-yl)-2,5-diphenyltetrazolium bromide (MTT) dye reduction assay was performed in 96-well microplates as previously described [36]. $3\times 10^{3}$ cells for BAEC and $5\times 10^{3}$ cells for HUVEC in a final volume of 100 μL of the medium with serial dilutions of DMI were incubated at 37 °C under a humidified 5% ${\mathrm{CO}}_{2}$ atmosphere. After 72 h, MTT was added, and the absorbance was read at 550 nm with an Eon Microplate Spectrophotometer from Bio-Tek Instruments (Winooski, VT, USA). Data were collected by Gen5 software from the same manufacturer. Half-maximal inhibitory concentration (IC_50_) values were estimated as the concentrations of the compound yielding a 50% cell number, taking the values obtained for the control condition (the cells treated with the vehicle, DMSO) to be 100%.

### 4.4. EdU Flow Cytometry Proliferation Assay

The baseclick EdU Flow Cytometry Kit (Merck, Darmstadt, Germany) was used to determine cell proliferation according to the manufacturer’s instructions. EdU (5-ethynyl-2′-deoxyuridine) is a nucleoside analog to thymidine that is incorporated into DNA during active DNA synthesis, and it is further detected upon a click-chemistry reaction. BAEC at 60% confluence were treated for 14 h with different doses of dimethyl itaconate and labeled with 10 μM EdU for 2 h before fixation. Next, the cells were permeabilized, treated with the assay cocktail for click-detection, and eventually analyzed in a FACS VERSETM flow cytometer (BD Biosciences, Franklin Lakes, NJ, USA). The Kaluza software was used for data analysis.

### 4.5. Endothelial Cell Tube Formation on Matrigel

Matrigel (50 μL of about 10 mg/mL per well) was polymerized at 37 °C for a minimum of 30 min. $4\times 10^{4}$ cells (BAEC) were added with 200 μL of medium without FBS with 0, 125, 250, or 500 μM DMI. An amount of 2 μM staurosporine was used as a positive control for the total inhibition of the tube formation [9]. After 5 h of incubation, cell cultures were photographed with a microscope camera Nikon DS-Ri2 coupled to a Nikon Eclipse Ti microscope from Nikon (Tokyo, Japan). Closed “tubular” structures were counted using FIJI software.

### 4.6. Cell Cycle Analysis by Flow Cytometry

BAECs were seeded in a 6-well plate and treated with 0, 125, 250, or 500 μM DMI or 10 μM 2-methoxyestradiol (2-ME) for 16 h (as a positive control) [9] during their exponential growth phase. The cells were harvested, washed with PBS with 1% FBS and HEPES (10 mM), and fixed with 70% ethanol for 1 h on ice. Finally, the cells were incubated with 0.1 mg/mL RNAse-A and 40 μg/mL propidium iodide for 1 h at 37 °C. The percentages of SubG1, G0/G1, and G2/S/M populations were obtained using a FACS VERSETM cytometer from BD Biosciences (San José, CA, USA) and analyzed using its software BD FACSuite.

### 4.7. “Wound Healing” Migration Assay

The migration of BAEC in the presence of DMI was assessed using the “wound healing” assay as previously described [26]. After the scratch, the wounded area for each condition (0, 125, 250, or 500 μM DMI) was photographed after 0, 4, and 8 h of incubation. Images were analyzed with FIJI software. The migratory capacity of DMI-treated BAEC was calculated as the percentage of the wounded area at time 0 h that had been recovered by the BAECs after the different incubation times.

### 4.8. Endothelial Cell Invasion Assay

The invasiveness of BAEC in the presence of DMI was assessed by using a 24-well clear membrane insert as described previously [37]. $5\times 10^{4}$ BAECs serum-starved were added to inserts coated with Matrigel in a serum-free media with 0, 125, 250, or 500 μM DMI. DMEM containing glucose (1 g/L) with 20% FBS was used as a chemoattractant in the lower wells. Serum-free medium was used as the negative control. After 16 h of incubation, the cells were fixed with 4% paraformaldehyde (PFA) and dyed with 1% violet crystal (on 2% ethanol). Finally, inserts were photographed, and migrated cells were counted with FIJI software. The percentage of cells migrated was calculated considering the number of cells in the positive control to be 100%.

### 4.9. Zymographic Assay for the Detection of Matrix Metalloproteinase-2 (MMP-2)

BAECs were incubated with media containing 0.1% BSA, 200 KIU/mL aprotinin, and 0, 125, 250, or 500 μM DMI for 16 h. Then, conditioned media were collected, centrifuged at 1000 g for 10 min at 4 °C, and used for gelatin zymography as previously described [36]. As positive controls, BAEC treatments with 20 µM toluquinol were used [10]. Duplicate samples were used to determine cell numbers with a Coulter counter. Gels were photographed with the ChemiDoc^TM^ XRS + System (Bio-Rad) using Image Lab^TM^ software. Quantitative analysis was performed using FIJI software.

### 4.10. Statistical Analysis

Replicates from at least three independent experiments were performed in all the assays. Quantitative results are expressed as means ± S.D. Statistical analysis of the obtained results was performed through a Student’s *t*-test with GraphPad Prism software, and *p* values < 0.05 were considered statistically significant. *p* values were represented as * *p* < 0.05, ** *p* < 0.01, *** *p* < 0.001, **** *p* < 0.0001 versus negative control.

## Figures and Tables

**Figure 1 ijms-23-15972-f001:**
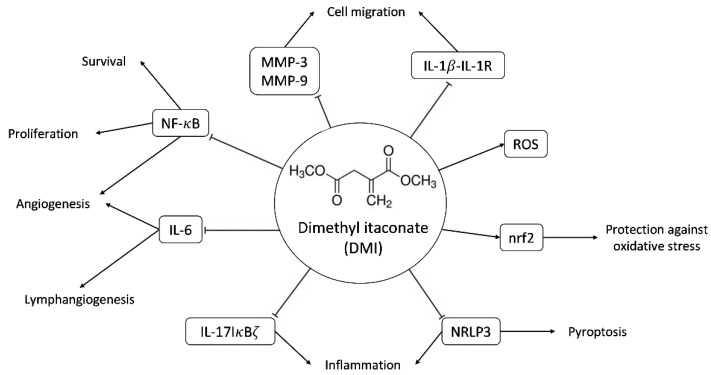
Scheme of some of the molecular targets of DMI and their biological effects. Abbreviations: matrix metalloproteinase-3 and -9 (MMP-3 and MMP-9), interleukin-1beta-interleukin 1 receptor (IL-1β-IL-1R), reactive oxygen species (ROS), nuclear factor-erythroid factor 2-related factor 2 (nrf2), nuclear factor kappa B (NF-κB), interleukin-6 (IL-6), interleukin-17 NF-kappa-B inhibitor zeta (IL-17-IκBζ), NLR family pyrin domain containing 3 (NLRP3).

**Figure 2 ijms-23-15972-f002:**
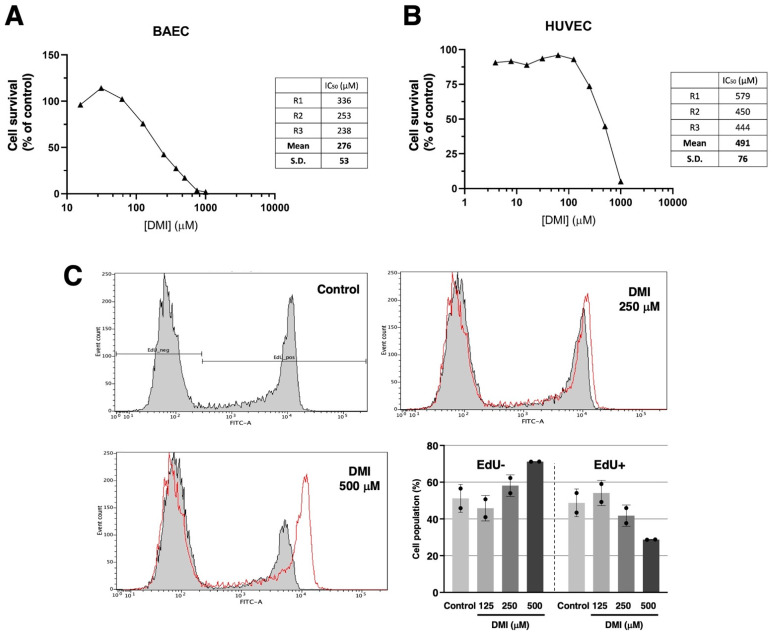
Effect of DMI on endothelial cell survival under proliferative conditions. Dose–response curves showing the effect of DMI on the in vitro growth of BAECs (**A**) and HUVECs (**B**) after 72 h treatment in low-density seeding conditions. Cell number is represented as the percentage of cells compared to the condition containing the vehicle, DMSO (at non-toxic concentrations, well below 1% *v*/*v*). Concentrations are represented in a logarithmic scale. Data from one representative experiment is shown. A table with the IC_50_ value obtained in each replicate and the media ± S.D. is included in each panel. (**C**) Flow cytometry representative profiles of 2 h EdU-treated BAEC, which were previously in the absence (control) or presence of DMI at different doses for 14 h, and the quantitative analysis of EdU− (non-proliferative) and EdU+ (proliferative) cells (media ± S.D. of two independent experiments).

**Figure 3 ijms-23-15972-f003:**
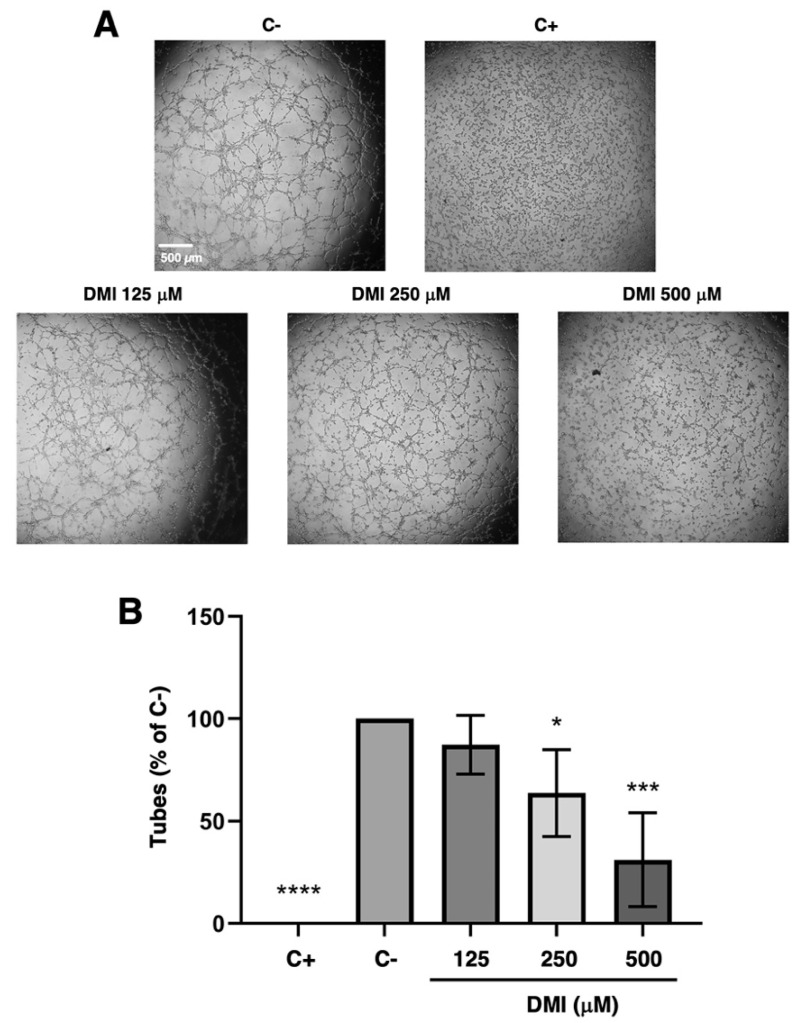
Effect of DMI on the formation of endothelial tubular structures. (**A**) Representative images of negative (DMSO, at non-toxic concentrations, well below 1% *v*/*v*) and positive control (2 µM staurosporine) [9] and DMI-treated BAECs on tube formation of Matrigel. Control cells formed tube-like structures. Cells were photographed 5 h after seeding and drug administration under an inverted microscope. (**B**) Quantitative analysis of “tubular” structures formed. Data are represented as means ± S.D. for four independent experiments. * *p* < 0.05, *** *p* < 0.001, **** *p* < 0.0001 versus negative control.

**Figure 4 ijms-23-15972-f004:**
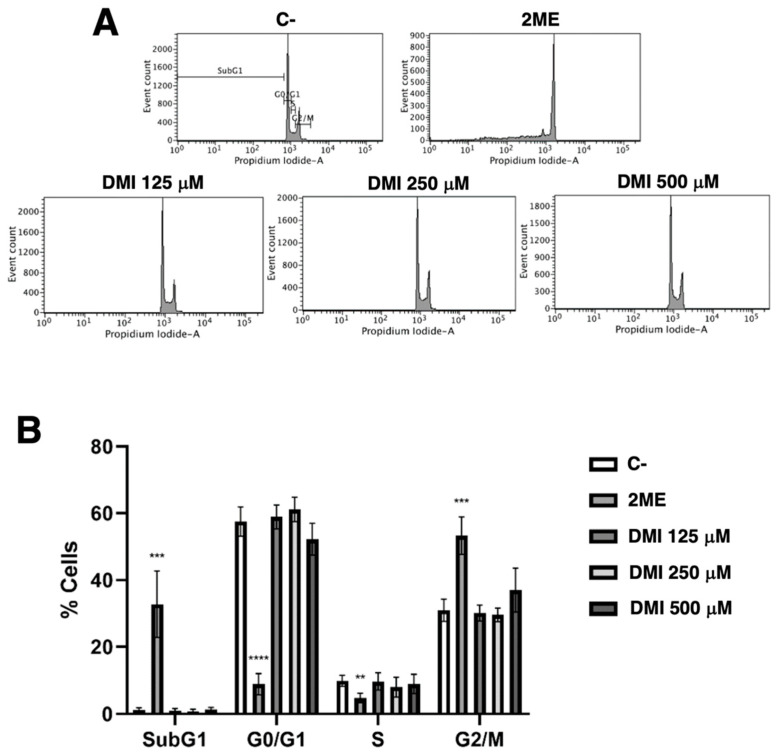
Effect of DMI on endothelial cell cycle. (**A**) BAECs were exposed for 16 h to DMI at the indicated concentrations to DMSO (negative control, at non-toxic concentrations, well below 1% *v*/*v*) or to 10 µM 2-ME (positive control) [9], stained with propidium iodide and the percentages of cells on subG1, G1, S, and G2/M phases were determined using a FACS VERSETM cytometer. (**B**) Quantification of the percentages of cells at each stage of the cell cycle. Data are represented as means ± S.D. for four independent experiments. ** *p* < 0.01, *** *p* < 0.001, **** *p* < 0.0001 versus negative control.

**Figure 5 ijms-23-15972-f005:**
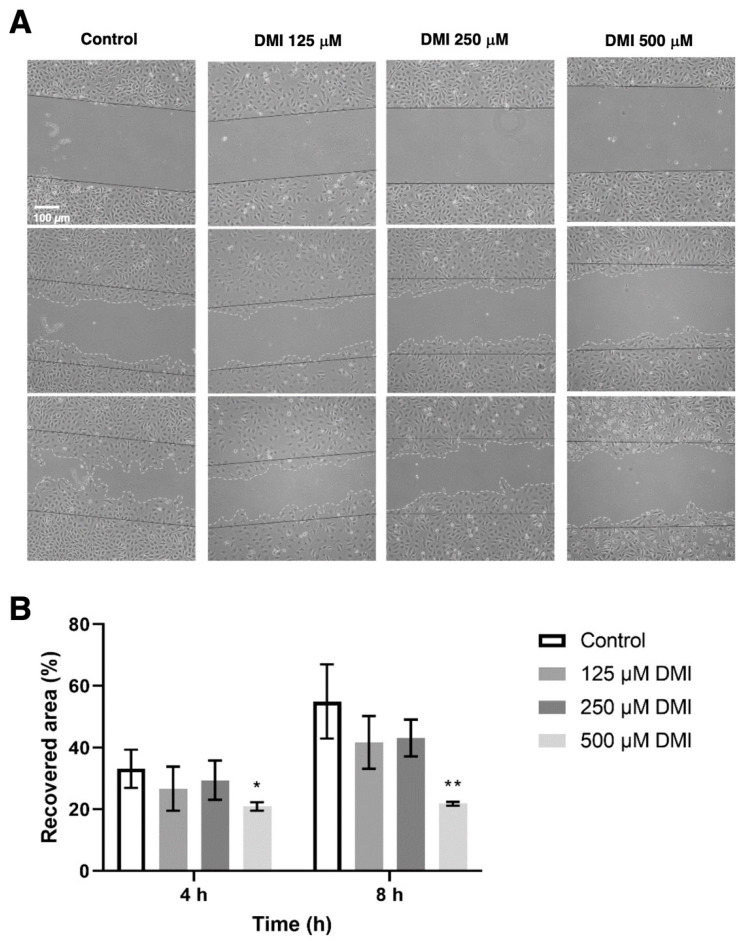
Effect of DMI on endothelial cell migration in vitro. (**A**) Representative images of control and DMI-treated BAECs at the indicated times of incubation. The initial area is indicated in the images with black continuous lines. The recovered area by the cells is indicated with grey dashed lines. (**B**) Quantification of the recovered area. Data are expressed as means ± S.D. of four independent experiments. * *p* < 0.05, ** *p* < 0.01 versus control.

**Figure 6 ijms-23-15972-f006:**
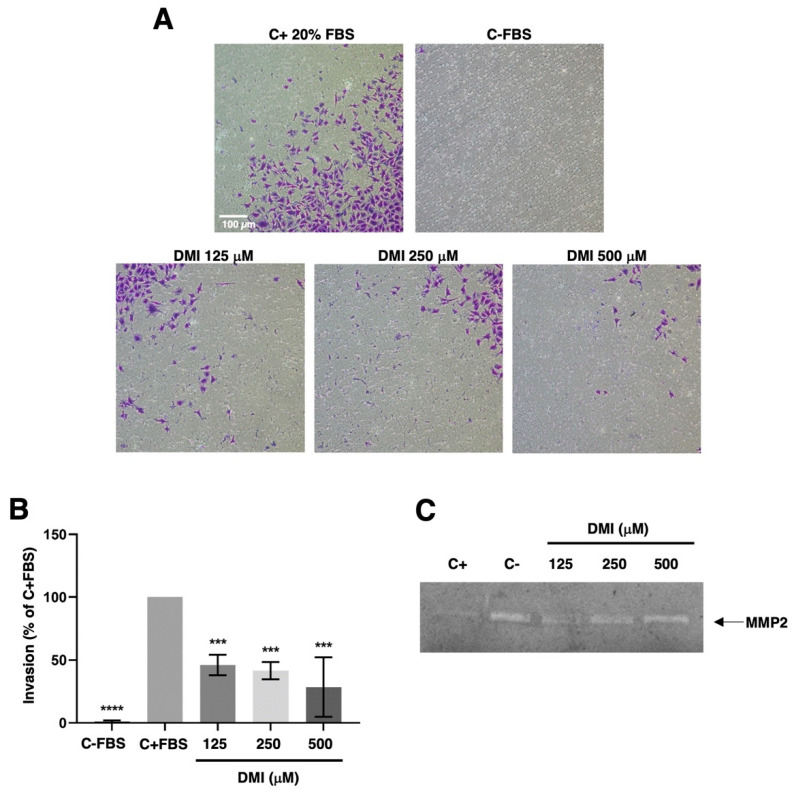
Effect of DMI on endothelial cell invasion and matrix metalloproteinase-2 secretion. (**A**) Representative photographs of positive and negative control and DMI-treated BAECs after 16 h of incubation in a transwell insert coated with Matrigel. FBS was added as a chemoattractant to the lower wells, except for the negative control. (**B**) Quantification of the results obtained in the invasion assay. Data are given as a percentage of the positive control and expressed as means ± S.D. of four independent experiments. *** *p* < 0.001, **** *p* < 0.0001 versus positive control. (**C**) Conditioned media from BAECs treated for 24 h with the indicated concentrations of DMI were normalized for equal cell density and used for gelatine zymography. A representative result from six independent experiments. The negative control is conditioned media from non-treated cells. The positive control is conditioned media from BAEC treated with 20 µM toluquinol [10].

## Data Availability

Not applicable.

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
