# Peer review of "The Immunomodulator Dimethyl Itaconate Inhibits Several Key Steps of Angiogenesis in Cultured Endothelial Cells"

_ijms, 2022, doi:10.3390/ijms232415972_

Round 1
Reviewer 1 Report
Isbel and Elena et al. investigate the anti-angiogenic potential of dimethyl itaconate utilizing cultured endothelial cells and in vitro assays such as endothelial cell proliferation, migration, invasion, and tube formations. Despite findings and substantial evidence, certain key findings must be supported; the mechanism is not obvious enough. Furthermore, based on the data, certain conclusions are not convincing.
Major comments:
1. Does DMI treatment inhibit cell growth or inhibit apoptosis? The MTT test is insufficient to conclude that it inhibits cell growth. Recommended showing cell morphologies for each concentration of DMI.
2. The authors demonstrated that DMI inhibits cell proliferation (growth); however, there is no significant effect on the cell cycle upon treatment with DMI (Fig4). The authors must clarify it.
3. The authors estimated the IC50 value for DMI using the MTT assay at 72 hours. The entire functional assay, however, was performed within 24 hours durations. For example, cell migration assay lasting 8 hours, 5 hr for tube formation assay, and a cell invasion assay lasting 16 hours. Is this small window of time sufficient to show the phenotype? The authors must verify the IC50 value in a time-dependent manner.
4. What is the mechanism through which DMI inhibits endothelial cell invasion? Are 16 hours long enough to notice a difference in MMP2 protein production with DMI treatment? The authors state that the IC50 value is 276 uM after 72 hours of incubation. Recommend seeing other metalloproteinases.
5. Figure 7; For the zymographic experiment, authors must include a positive and negative control.
6. The quantification graph (fig 6B) and the number of cells from the microscopic pictures (fig 6A) do not match. Figure 6A looks like DMI fully inhibits cell invasion (less than 5%). However, the quantification graph indicated that it was close to 50%.
7. An in vivo study is recommended.
8. Nowhere mention of Staurosporine or 2-ME role in the experiment? Brief information is required in the article.
9. Wondering to whom authors are thanking? See sentence number 130-131 “ ….angiogenesis is achieve thanks to the production…..”. Please rewrite the sentence.
10. The scale bar for figures 3A, 5A, and 6A must be included.
Minor comments:
1. Provide the citation for sentences 48 through 52.
2. Page 94 has to be corrected...” flow cytometer assay described….”
Author Response
Please, see the attachment

Reviewer 2 Report
Overall, the results support the conclusions. The introduction can be improved by adding specifics about mechanisms of action and in vivo studies in the context of inhibition of in vivo angiogenesis by the inhibitor. Sma for the Discussion. Figure 7 needs better description. The concentration of DMSO is also important and needs to be added in the figure legends. The paper would also improve if the compound was tested in an in vivo model of angiogenesis was tested. If this is not available, please discuss the dose, pharmacokinetics, and side effects of the use of this compound in other models of cancer and angiogenesis in vivo.
Author Response
Please, see the attached pdf document.

Round 2
Reviewer 1 Report
The manuscript looks fine and and the authors have addressed my concerns.
Author Response
Thank you for your comments and your overall positive evaluation.